# Pavement Performance and Ice-Melting Characteristics of Asphalt Mixtures Incorporating Slow-Release Deicing Agent

Jiaqiang Zhang [1], Weicheng Wang [2], Jinzhou Liu [1], Shuyi Wang [3], Xiaochun Qin [4] and Bin Yu [1],*

1  School of Transportation, Southeast University, Nanjing 210089, China
2  China Design Group Co., Ltd., Nanjing 210014, China
3  College of Civil Engineering, Fuzhou University, Fuzhou 350116, China
4  School of Civil Engineering, Beijing Jiaotong University, Beijing 100044, China
*  Correspondence: yb@seu.edu.cn

**Abstract:** Icy pavement is one of the primary causes affecting driving safety in winter, and deicing asphalt mixture could effectively resist pavement icing. This study evaluated the effect of a slow-release deicing agent on pavement performance and ice-melting characteristics of the asphalt mixture. The asphalt mixture containing four different contents (0%, 30%, 50%, and 70% using the internal mixing method) by replacing mineral filler was designed. Pavement performance tests were used to investigate the effect of the deicing agent on the high-temperature stability, cracking resistance, and water stability of the asphalt mixture. Qualitative and quantitative tests were designed to compare the ice-melting characteristics and predict the ice-melting durability with different replacement amounts. The experimental results show that the high-temperature stability, low-temperature cracking resistance, and water stability of the asphalt mixture decrease with the increase in the deicing agent. The mineral filler with a content of 50% deicing agent will enhance the stability of the mixture in the short term. Deicing asphalt mixture can significantly improve the ice and snow melting ability of the pavement, and the asphalt mixture with a content of 50% deicing agent will reduce the interface adhesion between mixture and ice by more than 55%. The slow-release deicing asphalt mixture can reach the maximum release concentration within two hours under rain and snow. The recommended replacement amount of slow-release deicing agent is 50%, and the predicted durability of deicing asphalt mixture is 5–8 years.

**Keywords:** road engineering; deicing mixture; pavement performance; ice-melting characteristics; durability prediction





## 1. Introduction

Snow and ice on asphalt pavement surfaces normally occur with snow or freezing rain weather in winter. A thin layer of ice will be created when snow and ice cover the pavement for a long time, leading to a reduction in the friction coefficient of the pavement surface and affecting the safety of vehicle driving [1]. Several methods have been proposed to remove the snow from the pavement surface, of which the common methods are mechanical plowing and manual sweeping. However, these kinds of methods require considerable specialized machinery and labor costs and may cause traffic congestion [2,3]. In addition, certain progress has also been made in active snow and ice removal methods, such as electrically conductive concrete, geothermal energy, microwave heating, and self-ice-melting asphalt pavement [4,5]. The self-ice-melting asphalt pavement is achieved by adding the deicing agent into the asphalt mixture. The active ingredient in deicing agents gradually migrating to the pavement surface under snow weather can reduce the freezing point, thus effectively reducing the icing situation of the pavement surface in winter to ensure traffic safety [6].

The deicing materials used for asphalt pavement were first studied as early as the 1960s. European countries such as Switzerland and Germany began developing deicing

agents, which consist of various salts, including sodium chloride and sodium acetate [7,8]. Based on the successful experience of deicing agents in other countries, Chinese researchers have conducted numerous experiments to develop inorganic halides and organic salt deicing agents, such as Icebane and ZGHIT low freezing point filler [9]. Nevertheless, there is no unified standard to guide the deicing performance test. In the past decades, many scholars have designed different tests to evaluate the melting performance of deicing agents. Xia et al. [10] evaluated the ice resistance of the deicing agent by pulling and shearing test and assessed the long-term deicing performance after long-term environmental simulation Zhao et al. [11] measured the mass of the melted ice on the Marshall specimen to calculate the ice-melting rate and evaluate the ice-melting ability of different specimens. Chen et al. [12] developed test instruments, including a pulling force tester and shearing force tester, which could reflect the ice-pavement adhesion by normal and horizontal adhesion strength. Giuliani et al. [13] recorded the water-freezing process on the surface of asphalt film with a camera and identified the freezing point value of the deicing asphalt mixture. It was concluded that deicing filler significantly delayed the ice formation on the pavement surface, accelerated the ice-melting speed, and reduced the adhesion between ice and pavement. Ma et al. [14] replaced 25%, 50%, 75%, and 100% of the mineral filler with a deicing agent and indicated that the deicing effect was more significant with the increasing content of the deicing agent, as shown by the reduction of ice fracture strength, peel strength, and shear strength on the surface of the mixture.

The asphalt mixture with a deicing agent could avoid the occurrence of pavement icing to a certain extent. However, the replacement of mineral filler with a deicing agent will jeopardize the performance of the pavement. Therefore, the effect of the deicing mixture on the pavement performance should be noted [15,16]. Many studies have been conducted on the pavement performance of anti-icing asphalt mixture. Han et al. [17] studied the pavement performance of three types of deicing agents, Mafilon, calcium acetate, and magnesium acetate, at different dosing levels. The results concluded that the replacement of mineral filler by deicing agents impaired the pavement performance of the asphalt mixture. Liu et al. [18] tested the low-temperature properties and salt-releasing characteristics of antifreeze asphalt concrete and indicated that such properties were affected by salt content. The continuous immersion improved the bending strain at early ages. Ma et al. [19] added a polyester fiber modifier to the deicing asphalt mixture and found that the above mixture possesses much better engineering properties than the normal one. Min et al. [20] compared the engineering performance of a styrene-butadiene-styrene (SBS) modified mixture and an epoxy asphalt mixture. The results showed that the snow-melting agent has a lower influence on the engineering performance of the epoxy asphalt mixture. Yu et al. [21] prepared the asphalt mixture with snowmelt agent and high-elastic binder, which proved favorable pavement and de-icing performance, and the mixture can remove an ice layer less than 12 mm thick.

Meanwhile, the deicing effect of asphalt mixture relies on the migration of active ingredients in the deicing agent. The active ingredient will be gradually lost during the melting process. Thus, deicing asphalt mixture has a certain effective life [22]. Several methods have been used to predict the service life of deicing asphalt mixture, including the ion electrode method and freezing point method [23,24]. The cumulative and single-release concentrations are used as indicators in the ion electrode method, combined with the concept of critical effective ion concentration to predict the service life of the deicing agent [25]. Zhang et al. [26] established the prediction model of deicing longevity of self-ice-melting asphalt pavement based on mathematical regression fitting. Wu et al. [27] used Mafilon to replace the mineral filler in the mixture and derived a linear relationship between the specific gravity of the salt solution and the freezing point according to the freezing point test, combined with the water immersion test to quantitatively predict the deicing durability. However, there are few studies on the durability of deicing asphalt mixture. No unified test method has been proposed to predict the durability of deicing

agents. Therefore, there remain issues in evaluating the pavement performance of the deicing mixture and the durability prediction of the deicing agent.

In conclusion, icy pavement surface is an important factor affecting driving safety in rain and snow weather. The active deicing technology of adding a deicing agent to the asphalt mixture can effectively solve this dilemma. A majority of the studies have been performed to evaluate the ice-melting ability and pavement performance of deicing asphalt mixture, while the corresponding behavior considering water stability requires to be clarified. Furthermore, the durability prediction of the deicing mixture is also significant. Therefore, this paper studied the pavement performance and melting characteristics of a slow-release deicing agent. The melting performance was evaluated in multiple dimensions by qualitative and quantitative methods. The immersion Marshall test after different immersion times and freeze-thaw splitting test were designed to verify the effect of the deicing agent on the water stability. Finally, the durability of the deicing agent was predicted based on the ice-melting ability. Figure 1 summarizes the experimental work and shows the flowchart of the methodology.

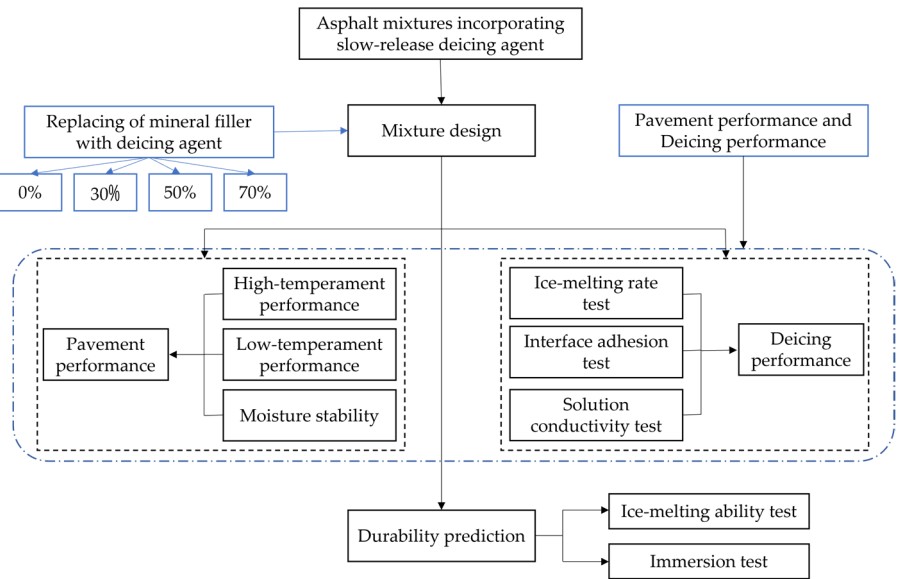

**Figure 1.** Experimental work flowchart.

## 2. Materials and Methods

### 2.1. Materials

Styrene-butadiene-styrene (SBS)-modified asphalt was used for this study. The basic properties of the SBS-modified asphalt are listed in Table 1. Basalt was used as the aggregate, and limestone powder was utilized as the mineral filler. The physical and mechanical characteristics of aggregate and mineral filler are shown in Table 2. It could be seen that all technical indices of asphalt and aggregate meet the requirements of the specification [28].

The deicing agent used in this study is ZGHIT, a product from Chenke Traffic Technology Co., Ltd. in Daqing, China, with a microscopic core-shell capsule structure of slow-release configuration. The maximum nominal particle size of the deicing agent is 0.075 mm, and the density is 2.20 g/cm$^3$, as shown in Figure 2. It consists of low surface energy water-repellent and low freezing point materials. The low freezing point materials could reduce the freezing point of the pavement, and the water-repellent materials could reduce water infiltration into the interior of the pavement, which can reduce the bonding ability of ice and pavement [11].

**Table 1.** Basic properties of SBS-modified asphalt.

| Test Items | Unit | Test Value | Requirements |
|---|---|---|---|
| Penetration (25 °C,100 g, 5 s) | 0.1 mm | 63 | 40–80 |
| Ductility (5 °C, 5 cm/min) | cm | 34.2 | ≥20 |
| Softening point | °C | 74.5 | ≥60 |
| Flash point | °C | 304 | ≥230 |
| Elastic recovery (25 °C) | % | 80 | ≥75 |
| Rolling thin film ovens test (RTFOT) | | | |
| Quality variation | % | 0.3 | ≤±1.0 |
| Penetration ratio (25 °C) | % | 76 | ≥65 |
| Ductility (5 °C) | cm | 23 | ≥15 |

**Table 2.** Engineering properties of aggregate with different particle sizes.

| Aggregates | 9–16 mm | 4.75–9.5 mm | 2.36–4.75 mm | 0–2.36 mm | Requirements |
|---|---|---|---|---|---|
| Apparent specific gravity | 2.862 | 2.879 | 2.846 | 2.787 | ≥2.60 |
| Bulk specific gravity | 2.713 | 2.722 | 2.709 | 2.683 | ≥2.50 |
| Water absorption (%) | 0.49 | 0.53 | 0.55 | 1.89 | ≤2.0 |

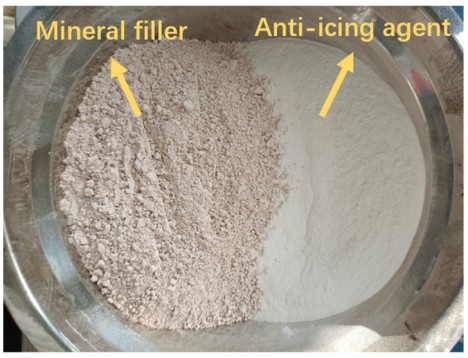

**Figure 2.** Mineral filler and ZGHIT deicing agent.

### 2.2. Mixture Design

The gradation of the deicing mixture was designed as a stone mastic asphalt (SMA) mixture with a maximum nominal particle size of 13 mm and a passing rate of 27.6% for the 4.75 mm sieve. The gradations of the mixture are illustrated in Figure 3. The lignin fiber with a content of 0.35% was blended in the mixture, and the optimum binder content was determined to be 6.2% by the standard [29]. ZGHIT deicing agent is a filler product, and it was incorporated into the mixture by internal mixing method, in which 0%, 30%, 50%, and 70% by volume of the mineral filler were replaced, respectively. The mixing process was the same as ordinary asphalt mixture, namely, adding aggregates and asphalt first and mixing for 90 s, then adding mineral filler and deicing agent for another mixing of 90 s.

### 2.3. Pavement Performance Tests

2.3.1. High-Temperament Performance Test

The high-temperature performance of the deicing asphalt mixture was evaluated via a wheel tracking test. The standard rutting specimen is a slab prepared by a roller compactor with a dimension of 300 mm × 300 mm × 50 mm according to the Chinese test specification [29]. Before the test, the rutting specimen was put into a constant temperature oven at 60 °C for more than 4 h. The test was carried out at 60 °C and a wheel pressure of 0.7 MPa. The dynamic stability (DS) was used to evaluate the high-temperature performance of the asphalt mixture, which was defined as the average loading cycles to form a 1 mm rutting depth. The number of specimens in each group was in accordance with the requirements of the specification, which is three parallel specimens [28,29]. If the error of a

certain group does not meet the requirements, the specimens will be prepared and tested again until they meet the requirements.

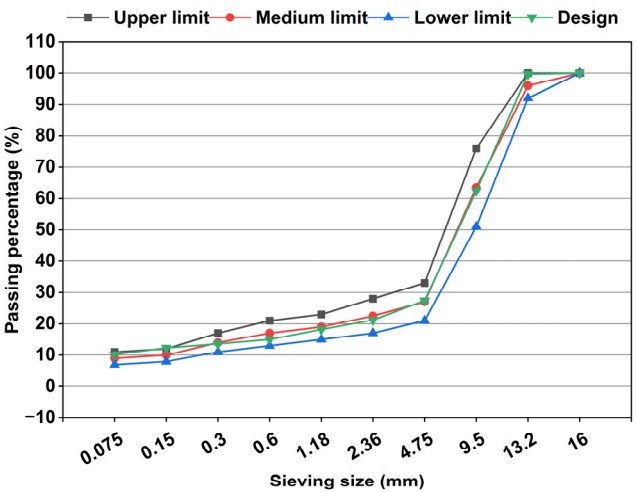

**Figure 3.** Grading curves of SMA-13 asphalt mixture.

### 2.3.2. Low-Temperament Performance Test

The low-temperature splitting test can determine the cracking resistance of the asphalt mixture at low temperatures. According to the test specification of the asphalt mixture splitting test in [29], firstly, the standard Marshall specimens ($\varnothing$101.6 mm $\times$ 63.5 mm) with different replacement amounts of the deicing agent were molded and then put into a $-10\,^{\circ}$C refrigerator for more than 4 h after demolding. Three parallel specimens were set in each group. During the splitting test, the specimens were loaded with a universal material testing machine at a constant rate of 1 mm/min until splitting damage under low-temperature conditions.

### 2.3.3. Moisture Stability Test

Moisture damage is one of the main forms of asphalt pavement distresses. The immersion Marshall test is usually used to assess the moisture damage resistance of the specimen. The specimens with different deicing agent replacement amounts were divided into two groups: the first group (MS_1) was tested for Marshall stability after 0.5 h in a water bath at $60 \pm 1\,^{\circ}$C; the second group (MS_2) was tested after 48 h in a water bath under the same conditions. In addition, the specification states that the residual stability after 48 h of water immersion normally meets the requirements in [30], while there may be additional cases where the residual stability is greater than 100%. Therefore, the Marshall stability of a constant temperature water bath for 10 days (MS_3) was tested for comparison.

Besides the immersion Marshall test, the indirect tensile strength ratio (TSR) test was also carried out to evaluate the moisture stability of the asphalt mixture under freezing and thawing conditions. According to the Chinese test specification [29], the test was performed via standard Marshall specimens, which were divided into two groups. The one group of specimens was maintained in a water bath at 25 $^{\circ}$C for 2 h, and then the indirect tensile strength $R_1$ was measured. The other group was first vacuum-saturated with water at 0.09 MPa for 15 min and then saturated with water at atmospheric pressure for 0.5 h. Then the samples were placed in a $-18\,^{\circ}$C refrigerator for 16 h and then immersed in a water bath at $60 \pm 0.5\,^{\circ}$C for 24 h, and finally immersed in water at 25 $^{\circ}$C for 2 h. After that, the indirect tensile strength $R_2$ was determined. The indirect tensile strength ratio was calculated by $R_2/R_1$. The number of parallel specimens used in the above tests was three.

*2.4. Deicing Performance Tests*

2.4.1. Ice-Melting Rate Test

The ice-melting test could be used to evaluate the ice-melting effect of deicing asphalt mixes visually. The Marshall specimens containing the deicing agent and 60 mL of the same size ice cylinders were prepared. The Marshall specimens were soaked in the water at a constant temperature of 5 °C to make them fully moistened and reduce the effect of water absorption on the ice melting rate of Marshall specimens. The ice cubes were placed over Marshall specimens, and the remaining mass of the ice cylinders was recorded at an interval of 10 min. In this test, the following parameter was obtained:

$$LR = \frac{m_0 - m_i}{m_i} \times 100 \qquad (1)$$

where *LR* is the ice melting rate, %; $m_0$ is the mass of the ice cylinder before melting, g; $m_i$ is the mass of the ice cylinder after melting weighed at *i* time, g.

2.4.2. Interface Adhesion Test

The interface adhesion test was designed regarding the adhesive binder bonding strength test (drawing method) in specification [29] to test the bonding force between the deicing asphalt mixture and the frozen sponge to reflect the de-icing effect of the deicing agent when the pavement is icy. The crossed nylon rope was fixed at the bottom of the sponge (120 mm × 100 mm × 60 mm) and then placed on the Marshall specimen after the sponge was filled with water. The combined specimen was then frozen in the refrigerator at −10 °C for 2 h until the water in the whole sponge was frozen into ice. The pointer-type tensiometer was hooked on one side of the nylon rope to slowly pull the frozen sponge and the Marshall specimen apart. Each group was set to three parallel specimens. The force during the pulling procedure was recorded as the interface adhesion, as shown in Figure 4. For the interface adhesion test, the smaller the pulling force between the icing sponge and the Marshall specimen means the weaker bond between the ice and the mixture.

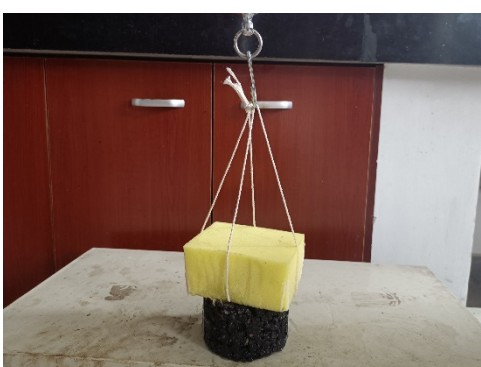

**Figure 4.** Interface adhesion test.

2.4.3. Solution Conductivity Test

The deicing performance of the slow-release deicing asphalt mixture depends on its salt compounds. After the Marshall specimens containing the deicing agent were placed in water, the deicing agent released ions that made the solution appear to have conductive [31]. Thus, the conductivity meter can be used to test the conductivity of the immersion solution to evaluate the deicing characteristics of the asphalt mixture indirectly. The test water was pure water, and the conductivity meter was CT-3031 ranging from 0 to 19.99 mS/cm.

*2.5. Durability Prediction of Deicing Mixture*

2.5.1. Ice-Melting Ability Test

The melting ability of the deicing agent is closely related to the concentration of the active ingredient. When the concentration of the active ingredient is lower than the critical concentration, the melting ability would be lost. According to the method of durability prediction of the deicing agent in [8], the ice-melting ability test of the solution with gradient concentration was designed to determine the critical concentration and the corresponding critical conductivity of the deicing agent. Nine groups of ice samples with thesame volume were prepared by filling a plastic cup with 50 mL of pure water and freezing them in a refrigerator. Correspondingly, nine solutions of different concentrations ranging from 0.8 to 4 g/L were prepared according to the concentration gradient of 0.4 g/L. After that the conductivity of each group was measured. Before starting the test, the solutions were placed in a 0 °C environment to reduce the impact of temperature on the test results, as shown in Figure 5. The solutions were subsequently poured onto the ice samples, and the volume of melted liquid was measured after 30 min. The melting volume of the ice sample was used to express the ice melting ability of the gradient concentration solutions.

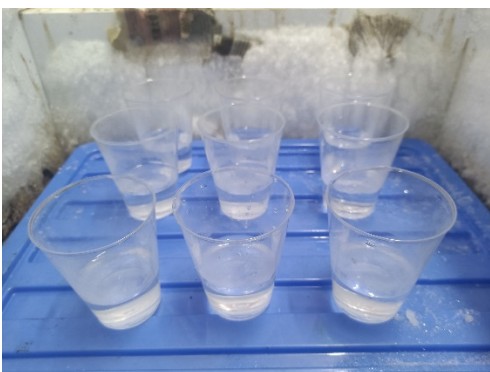

**Figure 5.** Ice-melting ability test of the solution with gradient concentration.

2.5.2. Immersion Test

The deicing components in asphalt pavement are released to the pavement surface through the connected pores in the mixture under the rainwater, thereby reducing the freezing point and playing the deicing role [32]. When the pavement structure is dry, it could be considered that the deicing agent is not released. At the same time, most of the rainwater flows into the rain well due to the road runoff in rain and snow weather, and the remaining rainwater penetrates the pores of the mixture under the action of external force, which could be approximately equivalent to the asphalt mixture immersed in water. Based on the above analysis and the results of the ice-melting ability test, the design of the immersion test used to predict ice-melting durability was as follows:

(1) Marshall specimen with 50% deicing agent content was placed in a container, and 600 mL pure water was injected to soak the top of the specimen slightly;
(2) After soaking for 24 h, the electrical conductivity of the solution was measured and recorded;
(3) The same volume of pure water was replaced and soaked again, and the cycle was repeated until the measured conductivity was lower than the critical value.

## 3. Results and Discussion

*3.1. Analysis of Pavement Performance*

3.1.1. High-Temperature Performance

As shown in Figure 6, the wheel tracking test results indicate that as the deicing agent content increases, the high-temperature performance of the asphalt mixture gradually decreases. The proportion of the decrease in DS is related to the amount of deicing agent

replacement. The DS values of asphalt mixture with 30%, 50%, and 70% deicing agent replacement decreased by 10.98%, 25.10%, and 30.10%, respectively compared to the mixture with 0% deicing agent, while the DSs of those mixtures are still able to meet the specification requirements in [30]. The decrease in high-temperature performance may be contributed to the fact that the specific surface area of deicing agent filler is smaller than that of mineral filler, which plays a role in absorbing excess asphalt in the asphalt mix. The replacement of mineral filler by the deicing agent will lead to the increment of free asphalt content and the corresponding reduction of structural asphalt content that provides bond strength and mechanical strength [33]. Therefore, the dynamic stability of the asphalt mixture decreases.

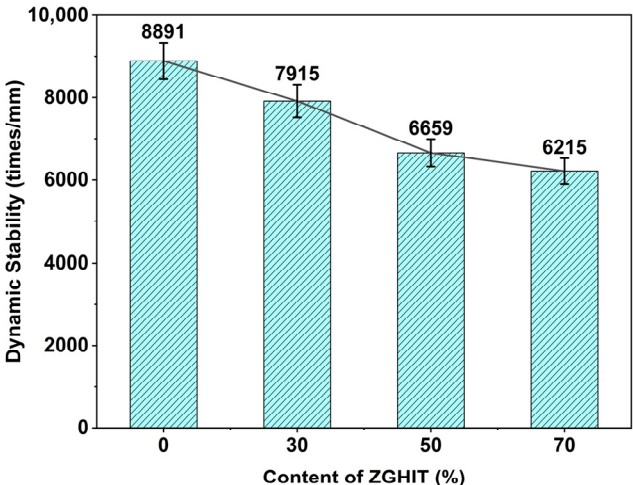

**Figure 6.** Results of wheel tracking test.

### 3.1.2. Low-Temperature Performance

Figure 7a shows the calculated splitting tensile strength of the deicing asphalt mixture, which reflects the stress required for splitting damage of the specimen. It was observed that replacing the mineral filler with the deicing agent would reduce the splitting tensile strength. The splitting tensile strength of the asphalt mixture without replacing the mineral filler is 3.64 MPa, while such strength with 30% replacement of deicing agent is 3.22 MPa, which is 11.6% lower than before. The line shows that the splitting tensile strength of asphalt mixture with a 50% replacement amount decreases by only 3.7% compared with that of 30%, and the splitting tensile strength of asphalt mixture with a 70% replacement amount decreases by 7.1% compared with that of 50%, which means that the splitting tensile strength of asphalt mixture with 50% deicing agent has a small decrease compared with that of the asphalt mixture with 30% deicing agent. In other words, the change in the content of the deicing agent from 30% to 50% has no evident influence on the low-temperature splitting tensile strength of the asphalt mixture. Figure 7b illustrates that the modulus of low-temperature failure strength of the asphalt mixture decreases with the increase inthe deicing agent content. This index is related to the flexibility of the asphalt mixture under low-temperature conditions. The increase in deicing agent replacement would lead to adecrease instructural asphalt in the mixture. In other words, the deicing agent would weaken the bonding effect between the aggregate and asphalt, which would reduce the low-temperature cracking resistance of the asphalt mixture [34].

### 3.1.3. Water Stability Performance

The results of Marshall stability values of asphalt mixture specimens with different soaking times are listed in Table 3. It could be seen that Marshall stability decreases with the increasing content of the deicing agent, and the residual stability of the mixture specimens with the deicing agent was higher than 100% after 48 h of soaking. The residual Marshall stability has a significant decrease when continues to soak up to 10 days. The reason is

that the moisture absorption capacity of salt-based deicing agents is stronger than that of mineral filler, and the active ingredients are less released to accommodate the expansion of the specimens after 48 h soaking, which makes the specimens more structured and have a certain increase in strength. Interestingly, the residual stability of conventional specimens did not change significantly with soaking time, while the residual stability of those with deicing agent behaves a significant decrease after 10 days in contrast to 48 h soaking. The migration of salt solution from high concentration to low concentration, on the one hand, changes the stable skeleton structure and increases the gap of the asphalt mixture. As a result, more water enters the asphalt mixture. On the other hand, the capillary effect during the migration process weakens the bond strength of asphalt and aggregate, and the final residual stability decreases significantly [18].

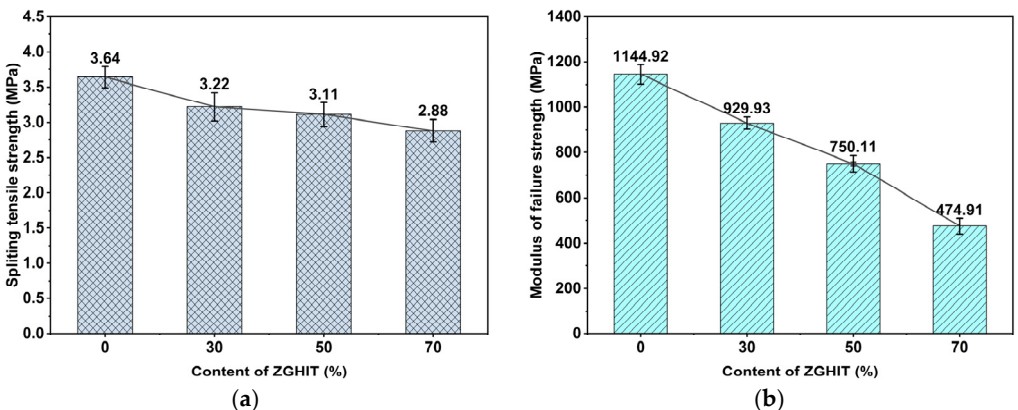

**Figure 7.** Results of the low-temperature splitting test. (**a**) Splitting tensile strength. (**b**) Modulus of failure strength.

**Table 3.** Results of water immersion Marshall test.

| Content of ZGHIT (%) | MS_1 (kN) | MS_2 (kN) | Residual Stability (%) | MS_3 (kN) | Residual Stability (%) |
|---|---|---|---|---|---|
| 0 | 16.63 | 16.15 | 97.11 | 13.88 | 88.46 |
| 30 | 12.76 | 15.59 | 122.18 | 8.35 | 65.43 |
| 50 | 12.62 | 15.20 | 120.44 | 7.26 | 57.52 |
| 70 | 11.81 | 14.11 | 111.95 | 6.67 | 56.48 |

As shown in Figure 8, the histogram indicates the splitting strength of the asphalt mixture without freeze-thaw and after freeze-thaw, and the line indicates the freeze-thaw splitting strength ratio of the corresponding asphalt mixtures. The splitting strength values of the specimens without freeze-thaw were approximately equal, while they show a decreasing trend with the increasing content of the deicing agent after freeze-thaw. The strength ratio of freeze-thaw splitting also shows the same trend. The results are considered as the active ingredients in the deicing agent would cause water intrusion, and the water entering the specimen would freeze and expand at low temperatures, which accelerates the speed of the asphalt detaching from the aggregate surface. In addition, during the freeze-thaw cycle, the precipitated salts may re-crystallize and expand, resulting in potential stress concentration on the void surface, which also aggravates the water damage of the specimen to a certain extent.

Based on the synthesis of the water immersion Marshall test and freeze-thaw splitting test, it can be concluded that the deicing agent replacement of mineral filler would reduce the water stability of the asphalt mixture, and the degree of water stability reduction is related to the replacement amount.

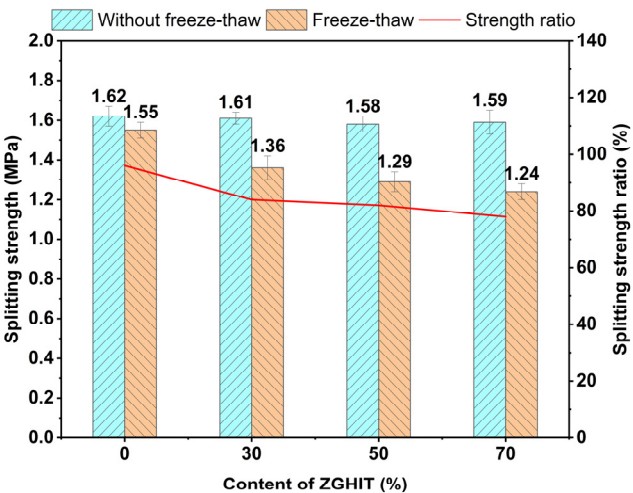

**Figure 8.** Results of the indirect tensile strength ratio test.

### 3.2. Analysis of Anti-Freezing Performance

### 3.2.1. Ice-Melting Rate

It could be visualized from Figure 9 that there was a significant difference in the ice melting effect of Marshall specimens with different deicing agent replacement amounts, and the deicing agent can improve the ice-melting ability of the asphalt mixture. The ice-melting rate of different specimens with time is shown in Table 4. When the melting time is 50 min, the specimens with 70% of the replacement amount of deicing agent melt all the ice, while the specimens without deicing agent melt only 18.28% of the ice. This suggests that the deicing agent has an obvious effect on melting ice. The ice-melting rates of the four groups of specimens were compared for 30 min, and it was found that the ice-melting rates of the specimens with 0, 30%, 50%, and 70% deicing agent replacement were 12.39%, 34.85%, 47.25%, and 63.34%, respectively, corresponding to the improvement of the ice melting performance of 1.81, 2.81 and 4.09. This means that the addition of a deicing agent can significantly improve the deicing ability of the specimens. However, the increase in the amount of deicing agent is limited to improve the ice-melting ability of the specimens.

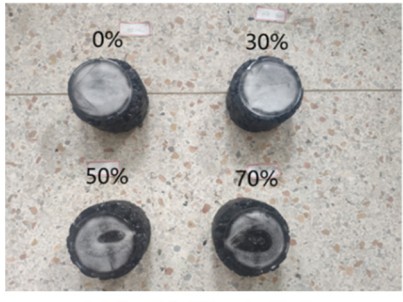
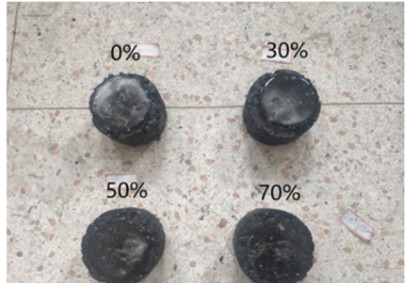

(a) Ice Melting in 5min　　　　　　　　(b) Ice Melting in 30min

**Figure 9.** Ice-melting rate of specimens with different content of deicing agent.

**Table 4.** Ice-melting rate of deicing specimens.

| Time (min) | | 0 | 10 | 20 | 30 | 40 | 50 |
|---|---|---|---|---|---|---|---|
| | 0 | 0 | 2.10% | 7.14% | 12.39% | 15.34% | 18.28% |
| Content of | 30 | 0 | 10.17% | 21.58% | 34.85% | 43.57% | 55.39% |
| ZGHIT (%) | 50 | 0 | 15.89% | 29.66% | 47.25% | 62.71% | 79.24% |
| | 70 | 0 | 21.26% | 44.03% | 63.34% | 86.55% | 100.00% |

### 3.2.2. Interface Adhesion

As mentioned above, the interface adhesion test could reflect the deicing effect of the deicing agent. The test results are shown in Figure 10. The bonding force between the asphalt mixture with the deicing agent and the frozen sponge is smaller than that of the conventional asphalt mixture. Moreover, the bonding force decreases gradually with the increase inthe replacement ratio of the deicing agent. Interestingly, the reduction of the bonding force with 30%, 50%, and 70% deicing agents is gradually smaller, with the reduction being 44.4%, 55.5%, and 61.1% compared with the conventional asphalt mixture. The interface adhesion is related to the release of the active ingredient of the deicing agent. The active ingredients are released to the pavement surface to reduce the freezing point and promote the melting of the ice layer. The smaller the bonding force means the weaker bond between the ice layer and pavement, which signifies the better melting effect of the deicing asphalt mixture. Similarly, the icing resistance is more evident after incorporating the deicing agent into the mixture, while it cannot be further improved if merely increasing the content of the deicing agent. This observation of the interface adhesion reduction agreed with the findings of the previous studies [19].

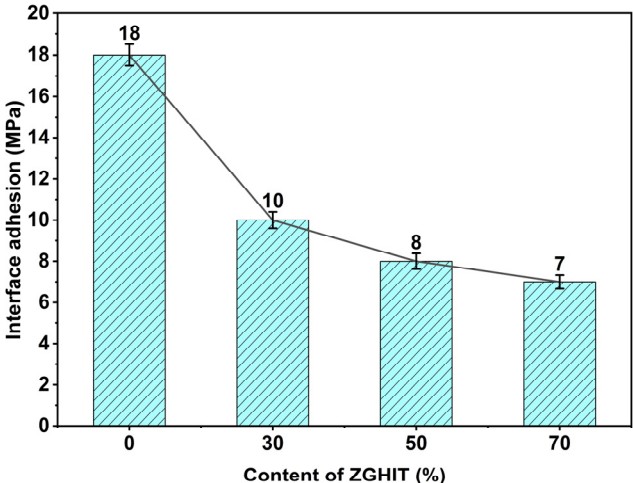

**Figure 10.** The bonding force between frozen sponge and deicing specimens.

### 3.2.3. Conductivity of Immersion Solution

Figure 11 demonstrates that the conductivity of the soaking solution after adding the deicing agent increases with the soaking time, especially within 2 h of soaking. The conductivity values of the soaking solution of different replacement mixtures are almost stable after 2 h. As a comparison, the conductivity of the solution of the conventional asphalt mixture maintains stability during the whole procedure. The conductivity curve of the soaking solution with 50% replacement of anti-icing agent asphalt mixture was between the conductivity curves of the soaking solution with 30% and 70% replacement of anti-icing agent asphalt mixture. Furthermore, the conductivity value of the soaking solution of asphalt mixture with 70% deicing agent replacement is about twice that of other deicing agent replacements of asphalt mixture. Undoubtedly, the conductivity of the soaked solution increases with the increasing content of the deicing agent.

According to the above results, the release rate of the slow-release deicing agent is positively correlated with its incorporating content in the mixture, and the deicing effect could be achieved more quickly with a larger replacement volume. Conductivity change with time could be divided into three stages including the rapid release stage, slow down release stage, and no longer release stage [17].

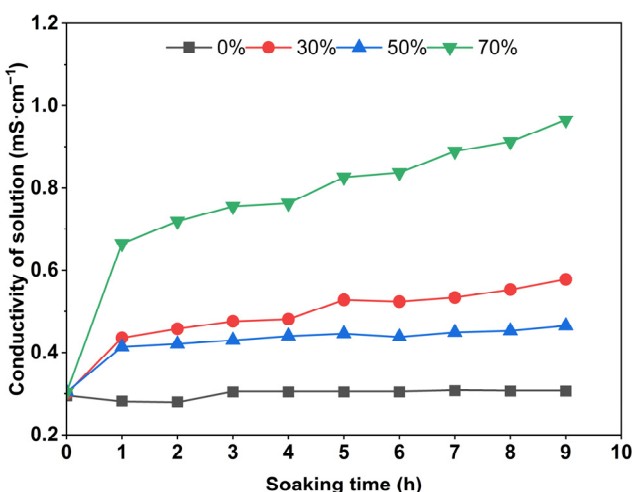

**Figure 11.** The release conductivity variation of deicing specimens at different times.

### 3.3. Durability Prediction

3.3.1. Ice-Melting Ability of Gradient Concentration Solution

Figure 12 shows the relationship between the melting amount of ice samples and the conductivity of the solution. It could be seen that there is a positive linear correlation between the concentration of the active ingredient solution and the conductivity, with a correlation coefficient greater than 0.99. The ice-melting ability increases with the increase inthe solution concentration. Furthermore, the ice-melting amount is approximately 0 when the solution concentration is 1.2 g/L, and the corresponding conductivity of the solution is 2.6 mS/cm. The freezing point of the solution needs to be reduced to make the ice sample melt, and the decrease inthe freezing point of the solution is related to the concentration of the active ingredient of the deicing agent. The ice sample would be melted only when the solution reaches a certain concentration, which could be defined as the critical ice-melting concentration [13]. Furthermore, the concentration of the solution could be indirectly characterized by measuring the conductivity. Therefore, it could be considered that the deicing mixture will lose the deicing capacity when the single-release concentration is lower than 1.2 g/L. That is, the solution conductivity is lower than 2.6 mS/cm. This critical ice-melting conductivity would be used to predict the durability of the deicing agent asphalt mixture in the immersion test.

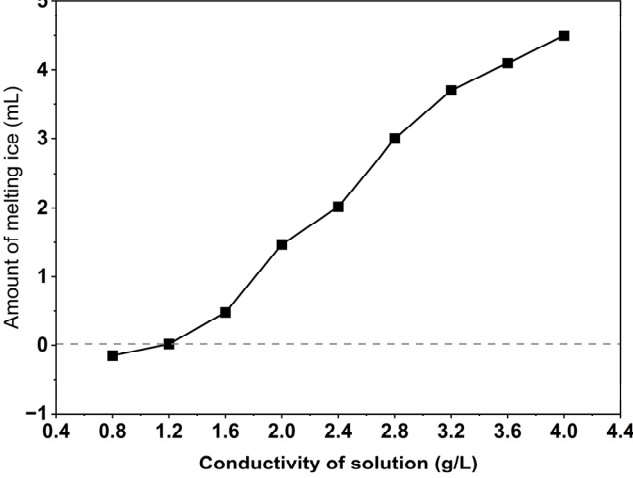

**Figure 12.** The amount of ice melting in the gradient concentration solution.

### 3.3.2. Durability Prediction of Deicing Agent

The durability prediction of the deicing agent was tested with an immersion test and, the experiment results are shown in Figure 13. It could be seen that the single-release concentration of the active ingredient in the Marshall specimen is gradually decreasing in all soaking cycles, and the changing rate in the conductivity of the solution at the beginning of soaking is low. This indicates that the single release of the deicing agent is generally stable and remains at a high level. At the middle stage of soaking, the decreased rate of conductivity increases obviously. This is due to the increasing porosity of the mixture resulting from the continuous release of the deicing agent, which in turn increases the precipitation rate of the active ingredients. By the end of the immersion cycle, most of the deicing filler in the mixture has been lost, resulting in a more stable decline in the rate of single releases. On the 15th day, the solution conductivity was down to the critical ice-melting conductivity of 2.6 mS/cm, at which point it is considered that the soaking sample has lost its deicing ability. The corresponding amount of water added to the test is 9000 mL.

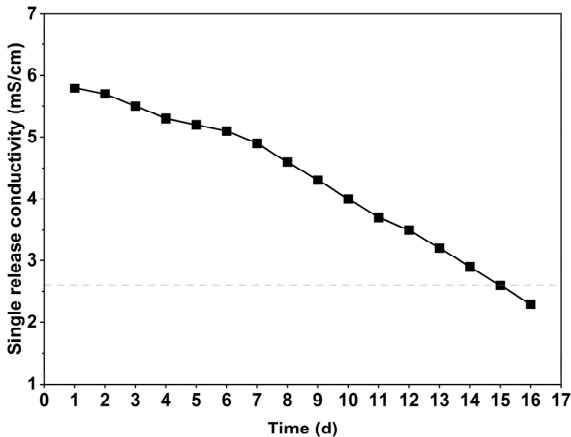

**Figure 13.** Cyclic soaking conductivity of deicing mixture.

The average annual rainfall data of five regions with different climate types in China were collected, and the runoff coefficient of urban asphalt pavement was considered to be 0.8. The conversion of average annual rainfall could be calculated by Equation (2), and the ice-melting durability of slow-release deicing asphalt mixture is calculated by equivalent conversion, as shown in Table 5. The service life of the deicing mixture has a great variation in different regions and is contrary to the value of the average annual rainfall. It can be calculated that the predicted life range is 5–8 years for various regions. In other words, the ice-melting ability of the asphalt mixture containing the deicing agent would be lost after exceeding the durability, and winter maintenance will be required.

$$V = \pi \times r^2 \times H \times (1 - a)/1000 \qquad (2)$$

where $V$ is the conversion of average annual rainfall, mL; $r$ is the radius of Marshall specimen, mm; $H$ is the average annual rainfall, mm; and $a$ is the runoff coefficient.

**Table 5.** Durability prediction of deicing agent in different regions.

| Region | Average Annual Rainfall (mm) | Conversion of Average Annual Rainfall (mL) | Durability Prediction (Year) |
|---|---|---|---|
| Heilongjiang | 753.1 | 1250.9 | 7.46 |
| Beijing | 644.2 | 1031.4 | 7.59 |
| Shaanxi | 740.4 | 1185.7 | 7.59 |
| Jiangsu | 1090.7 | 1746.8 | 5.15 |

### 4. Conclusions

To explore the effectiveness of adding a deicing agent to the asphalt mixture, the influence of the deicing agent on pavement performance and ice-melting characteristics were studied. Furthermore, the durability of the deicing asphalt mixture was predicted. Based on the entire investigation, the following conclusions can be summarized.

(1) The high-temperature performance and low-temperature cracking resistance of the asphalt mixture continuously decrease with the increasing content of the deicing agent. The replacement of mineral filler with a deicing agent can enhance the Marshall stability of the mixture in the short term, while the long-term residual stability after the release of the active ingredient is lower than that of the ordinary mixture.

(2) Deicing asphalt mixture can significantly improve the melting and snow removal ability of pavement. The interface adhesion can be reduced by more than 55% with a 50% replacement quantity. Considering the deicing performance and pavement performance of the mixture, the recommended replacement amount of slow-release deicing agent is found to be 50%.

(3) The slow-release deicing asphalt mixture can reach the maximum release concentration in 2 h under rain and snow, after which the solution maintains a stable concentration.

(4) The ice-melting capacity of the slow-release deicing agent is positively correlated with the released concentration. The effective life span of the deicing agent for this study is estimated to be 5–8 years after equivalent conversion.

Nonetheless, we also found that due to the lower surface area of the deicing agent than that of mineral fillers, adjusting the binder content of asphalt mixtures with different deicing agent contents might be beneficial to improving the pavement performance of the mixtures. It seems necessary to ensure the pavement performance of asphalt mixture containing the deicing agent and it is a promising research topic in the future. The influence of different binder content on the asphalt mixture with the deicing agent is also worthy of exploration.

**Author Contributions:** Conceptualization, J.Z. and B.Y.; Data curation, J.Z., W.W., J.L. and S.W.; Formal analysis, J.Z., W.W., X.Q. and S.W.; Funding acquisition, B.Y. and X.Q.; Methodology, J.Z.; Resources, W.W. and B.Y.; Validation, J.Z. and J.L.; Writing—original draft, J.Z.; Writing—review & editing, J.Z., J.L., X.Q. and S.W. All authors have read and agreed to the published version of the manuscript.

**Funding:** This research was funded by the Postgraduate Research & Practice Innovation Program of Jiangsu Province, grant number SJCX21_0057 and the Natural Science Foundation of China, grant numbers 51878163, 51878039 and 52078034.

**Data Availability Statement:** Not applicable.

**Conflicts of Interest:** The authors declare no conflict of interest.

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
