# Peer review of "Pavement Performance and Ice-Melting Characteristics of Asphalt Mixtures Incorporating Slow-Release Deicing Agent"

_buildings, doi:10.3390/buildings13020306_

Round 1

Reviewer 1 Report

The paper presents the study of mechanical and deicing properties of asphalt mixtures with different percentages of deicing agent.

The performance of asphalt mixture in general decreases with the increase of content of deicing agent. The authors claim that due to lower surface area of deicing agent, there is surplus of bitumen in the mixture. The question is why authors didn’t optimize/lower percentage of bitumen in mixtures containing deicing agent.? On that way more realistic comparison of performance between different mixtures would be possible.

Regarding low-temperature performance authors claim that for mixture with 50% of deicing agent that there was no evident effect of impact of deicing agent on performance. However, the decreasing trend with increasing content of agent is clearly present, and this statement can hardly be justified. It is necessary to specify the number of specimens used for all tests presented in the paper.

The paper states that durability of the asphalt mixture with deicing agent is 5 to 8 years, which means that surface layer should be replaced after this period. The question is how the cost of surface layer replacement compares to the reduction of cost of winter maintenance when asphalt mixtures with deicing agents are used.

Author Response

The authors (“we” hereafter) thank the anonymous reviewers very much for their valuable comments and suggestions and for their time in helping us improve the quality of this manuscript. We have improved both the content and composition of the revised manuscript substantially, following the comments and suggestions from the reviewers. Our point-by-point responses are detailed in this report. For clarity of presentation, our responses are presented in ‘Track Changes’ function.

Point 1: The paper presents the study of mechanical and deicing properties of asphalt mixtures with different percentages of deicing agent.

The performance of asphalt mixture in general decreases with the increase of content of deicing agent. The authors claim that due to lower surface area of deicing agent, there is surplus of bitumen in the mixture. The question is why authors didn’t optimize/lower percentage of bitumen in mixtures containing deicing agent.? On that way more realistic comparison of performance between different mixtures would be possible.

Response 1: We sincerely appreciate your review and good comments. The main purpose of this paper was to study the influence of different deicing agent content on pavement performance and ice-melting characteristics of asphalt mixture, while such content was not the focus of this research. In this study, Marshall test was used to determine the optimum binder content of asphalt mixture without deicing agent, and then its value was used to determine the optimum binder content of asphalt mixture with different content of deicing agent. In the test results of high-temperature performance, we found that the increase of deicing agent content would lead to the decrease of high-temperature performance of asphalt mixture. When trying to explain this reason, we found that the references [33] provided ideas, so it was cited as a further explanation and proof of the results. Just as the reviewer’s question, when the content of deicing agent in asphalt mixture increases, what effect would lower binder content have on the high-temperature performance of asphalt mixture? We think this provides potential orientation for our next research, and in the follow-up study, we will devote our efforts to studying the influence of binder content on the performance of asphalt mixture with different content of deicing agent.

Modification 1: We have supplemented suggestions to carry out research on the influence of binder content on the performance of asphalt mixture with different content of deicing agent at the end of the Conclusions in the revised manuscript (lines 467 to 473).

Point 2: Regarding low-temperature performance authors claim that for mixture with 50% of deicing agent that there was no evident effect of impact of deicing agent on performance. However, the decreasing trend with increasing content of agent is clearly present, and this statement can hardly be justified.

Response 2: We thank the reviewer for the question. As described in the result, the low-temperature performance of asphalt mixture decreases gradually with the increase of deicing agent content in asphalt mixture. The sentence you pointed out is a little unclear in our description. What we want to express is that the splitting tensile strength of asphalt mixture with 50% deicing agent has a small decrease com-pared with that of the asphalt mixture with 30% deicing agent. In other words, the change of the content of deicing agent from 30% to 50% has not evident influence on the low-temperature splitting tensile strength of asphalt mixture. This result is not to compare the low-temperature performance of asphalt mixture containing 50% deicing agent and the asphalt mixture without deicing agent.

Modification 2: We have rephrased the sentence to make the result more accurate in the revised manuscript (lines 288 to 292).

Point 3: It is necessary to specify the number of specimens used for all tests presented in the paper.

Response 3: We thank the reviewer for the suggestion. We have supplemented the number of specimens used for tests in methods section.

Modification 3: We have supplemented the number of specimens in the revised manuscript.

Point 4: The paper states that durability of the asphalt mixture with deicing agent is 5 to 8 years, which means that surface layer should be replaced after this period. The question is how the cost of surface layer replacement compares to the reduction of cost of winter maintenance when asphalt mixtures with deicing agents are used.

Response 4: We thank the reviewer for the question. The durability of the asphalt mixture containing deicing agent is predicted to be 5 to 8 years, which means that the ice-melting properties of the asphalt mixture will be lost after this period, rather than the need to replace the surface layer. The deicing agent used in this study has a slow-release configuration with a microscopic core-shell capsule structure. The effective components are adsorbed in porous supports, and the surface of the supports is coated with a hydrophobic material, so that the ingredient can slowly release to achieve a long-term snowmelt effect [31]. After the deicing component is released, the ice-melting ability of asphalt pavement is lost, but its mechanical properties can still meet the requirements of use. In other words, asphalt mixture containing deicing agent could still be used after exceeding the durability, then need winter maintenance on the asphalt pavement. Therefore, we think it is beneficial to use asphalt mixture with deicng agent in areas prone to pavement icing.

Modification 4: We have modified the results of durability prediction of deicing agent to express our results more clearly in the revised manuscript (lines 435 to 437).

Reviewer 2 Report

The article is interested. However, a major revise is suggested

1.     Abstract: line 9-13 should be revised

2.     Several grammatical errors were observed, please rectify the mistakes

3.     Flow chart is suggested to be added

4.     The error bars should be included

5.     Please add the standard specifications that you followed when doing the TSR test

6.     What is standard test related to the Interface adhesion test? Please include it

7.     Ice-melting ability test need to be supported by references

8.     Please delete figures 3 and 4

9.     The resolution of the figures should be improved

10.  The results in general need to be justified with references. This will support your achievement.

11.  The explanations associated with sub-heading 3.2.2, 3.2.3 and 3.3.1 are not sufficient, and should be enhanced

12.  The conclusion should be condensed and concise

Author Response

The authors (“we” hereafter) thank the anonymous reviewers very much for their valuable comments and suggestions and for their time in helping us improve the quality of this manuscript. We have improved both the content and composition of the revised manuscript substantially, following the comments and suggestions from the reviewers. Our responses are detailed in this report. For clarity of presentation, our responses are presented in ‘Track Changes’ function.

Point 1: The article is interested. However, a major revise is suggested

Abstract: line 9-13 should be revised

Response 1: We sincerely appreciate your review and good comments. The abstract (lines 9 to 13) has been revised.

Modification 1: We have revised the abstract (lines 9 to 15) in the revised manuscript.

Point 2: Several grammatical errors were observed, please rectify the mistakes.

Response 2: We thank the reviewer for the careful review. The manuscript has been thoroughly proofread to improve the language and to avoid grammar errors, and unclear descriptions.

Modification 2: We have gone over the grammar and made some corrections in the revised manuscript.

Point 3: Flow chart is suggested to be added.

Response 3: We thank the reviewer for the suggestion. The flowchart has already been supplemented in the Introduction section (lines 110 to 113).

Modification 3: We have supplemented the flowchart in the revised manuscript.

Point 4: The error bars should be included.

Response 4: We thank the reviewer for the suggestion. We have redrawn the figures and added error bars. Moreover, we reprinted the figures at a higher resolution.

Modification 4: We added error bars in the figures, the new figures could be seen in the revised manuscript.

Point 5: Please add the standard specifications that you followed when doing the TSR test.

Response 5: We thank the reviewer for the suggestion and agree with the reviewer. The standard specification that we followed about TSR test is Standard Test Methods of Bitumen and Bituminous Mixtures for Highway Engineering (JTG E20-2019).

Modification 5: The standard specification of TSR test has been supplemented in the revised manuscript (line 183).

Point 6: What is standard test related to the Interface adhesion test? Please include it.

Response 6: We thank the reviewer for the suggestion. The design of the interface adhesion test referred to the adhesive binder bonding strength test (drawing method, T0678) in the specification [29]. Based on this method, we made corresponding improvements according to the test conditions and purposes. For the interface adhesion test, smaller the pulling force between the icing sponge and the Marshall specimen means the weaker bond between the ice and the mixture.

Modification 6: We have supplemented the design idea of interface adhesion test in more detail and attached the reference. Furthermore, we have given a further explanation of the interface adhesion test.

Point 7: Ice-melting ability test need to be supported by references.

Response 7: We thank the reviewer for the suggestion. The ice-melting ability test was designed based on the method of durability prediction of deicing agent in reference [8], and we made some improvements on this basis.

Modification 7: We have supplemented reference to the section of ice-melting ability test, which used to support the experimental design (lines 232 to 233).

Point 8: Please delete figures 3 and 4.

Response 8: We thank the reviewer for the suggestion. The figures 3 and 4 have been deleted.

Modification 8: We have removed the figures 3 and 4.

Point 9: The resolution of the figures should be improved.

Response 9: We thank the reviewer for the suggestion. The resolution of the figures has been improved.

Modification 9: We have improved the resolution of figures in the revised manuscript.

Point 10: The results in general need to be justified with references. This will support your achievement.

Response 10: We thank the reviewer for the suggestion. We have supplemented the corresponding references in the results to further support the achievement.

Modification 10: We have supplemented the reference and revised the corresponding content in the revised manuscript.

Point 11: The explanations associated with sub-heading 3.2.2, 3.2.3 and 3.3.1 are not sufficient, and should be enhanced.

Response 11: We thank the reviewer for the suggestion. We have supplemented more explanations and discussions in light of previous literature to enhance the results in sub-heading 3.2.2, 3.2.3 and 3.3.1.

Modification 11: We have supplemented more explanations and discussions in light of previous literature in the revised manuscript.

Point 12: The conclusion should be condensed and concise.

Response 12: We thank the reviewer for the suggestion. We have revised the conclusion to be condensed and concise.

Modification 12: The conclusion has been revised in the revised manuscript (lines 423 to 443).

Round 2

Reviewer 2 Report

Thank you for rectifying the defects